# TF-HOT: Training-Free Hand-Object Pose Tracking and Optimization for Dexterous Manipulation

## Abstract

Robotic manipulation with dexterous hands is inherently challenging due to their high-dimensional action spaces and the lack of large-scale, high-quality demonstrations. While there are many videos involving interactions between human hands and objects, the frequent, dynamic occlusions between human hands and objects complicate the accurate and robust tracking of hand and object poses, making it challenging to convert these interactions into high-quality dexterous robotic demonstrations. To address these challenges, we introduce a novel Training-Free Hand-Object pose tracking pipeline (TF-HOT) that leverages differentiable rendering and rich priors from pre-trained 2D foundation perception models to perform optimization of human hand and object pose trajectories from input videos. Our method is efficient, allowing us to convert an in-the-wild video to pose trajectories in 1 minute, and we demonstrate state-of-the-art performance of our method over in-the-wild videos. Finally, we illustrate an application of our method in imitation learning by training policies to follow the pose trajectories extracted from TF-HOT, allowing us to learn dexterous manipulation policies that significantly outperform reinforcement learning and imitation learning methods that do not utilize hand-object pose trajectory following.

## 1 Introduction

Dexterous hand manipulation tasks are crucial in advancing robotics and artificial intelligence, with significant applications in areas such as virtual reality (VR), teleoperation, and human-robot interaction (Qin et al., 2022). These tasks are inherently challenging due to their high-dimensional state and action spaces, which lead to poor sample efficiency and complex reward designs when using reinforcement learning (RL) methods (Chen et al., 2022; Wang et al., 2024). Recent advancements in imitation learning have provided alternative approaches to tackle these challenges by leveraging demonstrations to guide the learning process (Qin et al., 2023). However, obtaining high-quality demonstrations for dexterous manipulation is non-trivial. Traditional methods often rely on devices like VR equipment or exoskeletons for teleoperation (Cheng et al., 2024; Yang et al., 2024b; Fang et al., 2024), which may not capture natural human hand movements and may be costly for scaling up to diverse environments and object interactions. Additionally, the discrepancy between human hand kinematics and robotic hand designs introduces gaps between simulation and reality.

Hand-object pose estimation is a fundamental task in computer vision (Chen et al., 2023; Qi et al., 2024; Liu et al., 2021; 2022; Yang et al., 2021; 2024a; Hasson et al., 2019) that can bridge this gap by extracting key information from human demonstrations captured in the wild. Accurate estimation of hand and object poses enables robots to learn manipulation skills directly from human behaviors observed in unstructured environments. However, there are several challenges: **1) Occlusion**: Severe occlusions during hand-object interactions degrade the performance of pose estimation algorithms. **2) Object diversity**: Existing datasets are limited in object diversity and are often captured in controlled environments, limiting the generalizability of trained models to unseen objects and in-the-wild scenarios. **3) Annotation difficulties**: Obtaining precise 3D annotations for hand and object poses is labor-intensive and impractical at scale, especially in real-world settings.

Current approaches to hand-object pose estimation can be broadly categorized into learning-based methods and optimization-based methods. Learning-based methods typically require datasets with object and hand pose annotations, which are expensive and time-consuming to collect at large scale, especially for real-world data. Thus, these models often struggle to generalize to unseen objects and complex environments. On the other hand, optimization-based methods can potentially generalize better but often rely on multi-camera setups, which limits their use in real-world scenarios.

To address these limitations, we introduce a novel, training-free hand-object pose tracking approach called **TF-HOT** (Training-Free Hand-Object Tracking). Our core idea is to perform inference-time optimization of pose parameters by utilizing both 3D point cloud observations and rich priors from pre-trained 2D foundation perception models. This method eliminates the need for large-scale annotated data for training a hand-pose tracking model, thereby reducing training costs and enhancing our method's adaptability for real-world applications. Specifically, we parameterize the optimization variables as a 3D parametric hand model (i.e., MANO (Romero et al., 2022)) and a 6DoF object pose. We then leverage 2D hand joints, 2D hand and object masks, 3D point cloud observations, and multiple regularization terms as constraints to guide the optimization process. This design effectively utilizes the rich knowledge and strong generalization capabilities of pre-trained 2D perception models, resulting in a robust and generalizable hand-object pose tracking system.

Our approach offers several key features: **Training-free deployment**: TF-HOT requires no model training. Each trajectory's hand-object pose optimization process can be completed within 1 minute, making TF-HOT adaptable to diverse scenarios and enabling scalable data generation for applications like imitation learning. **High accuracy and robustness**: By utilizing differentiable rendering-based joint optimization of hand and object poses and incorporating rich semi-supervisory signals from existing 2D foundation models, TF-HOT achieves high accuracy across diverse environments and remains robust to hand-object occlusions. We also demonstrate the superior performance of TF-HOT on the public DexYCB dataset compared to baseline methods. For imitation learning applications, we further propose a method named **Pose Trajectory Following (PTF)** that trains policies to control the robot to follow the pose demonstration trajectories extracted by TF-HOT. By doing so, we can effectively learn dexterous manipulation tasks and significantly outperform imitation learning and reinforcement learning methods that do not utilize hand-object pose trajectory following.

## 2  RELATED WORKS

**Hand and Object Pose Estimation**   Hand pose estimation has been tackled using various input modalities, including depth-based, RGB-based, and multimodal approaches. Depth-based methods have leveraged Principal Component Analysis (PCA), and convolutional neural networks (CNNs) in works such as (Oberweger et al., 2015; Oberweger & Lepetit, 2017). Additional CNN-based approaches have been proposed, including (Tompson et al., 2014) and (Madadi et al., 2017), while (Malik et al., 2018) introduced synthetic depth data for training. Ensemble learning strategies were also employed (Guo et al., 2017b;a) and are used in some anchor-based methods (Xiong et al., 2019). A recurrent approach using LSTM was presented in (Deng et al., 2022). For RGB-based methods, (Ge et al., 2019) proposed a Graph CNN, while (Zimmermann & Brox, 2017) introduced network-implicit 3D articulation priors. Works such as (Jiang et al., 2023) utilized 3D anchor points. Furthermore, methods extending MANO (Romero et al., 2022), incorporating partial depth information, were explored by (Baek et al., 2019).

In terms of object pose estimation, direct inference of 3D poses has been addressed through various approaches (Xiang et al., 2017). Two-step methods, which first lift 2D keypoints to 3D, have been explored in works such as (Tekin et al., 2018) and (Kehl et al., 2017), while coarse-to-fine strategies were employed in (Rad & Lepetit, 2017). Object pose estimation under severe occlusion has also been investigated by (Peng et al., 2019). (Sun et al., 2022) introduced a non-CAD-based method for real-time pose tracking. CAD-based methods like (Wen et al., 2024) have shown limitations when dealing with heavily occluded objects.

Many hand-object interaction methods are based on the MANO model (Liu et al., 2021; Chen et al., 2023), and several extended approaches have been proposed to improve performance. These extensions include the use of contact potential fields (Yang et al., 2021; 2024a), biomechanical con-

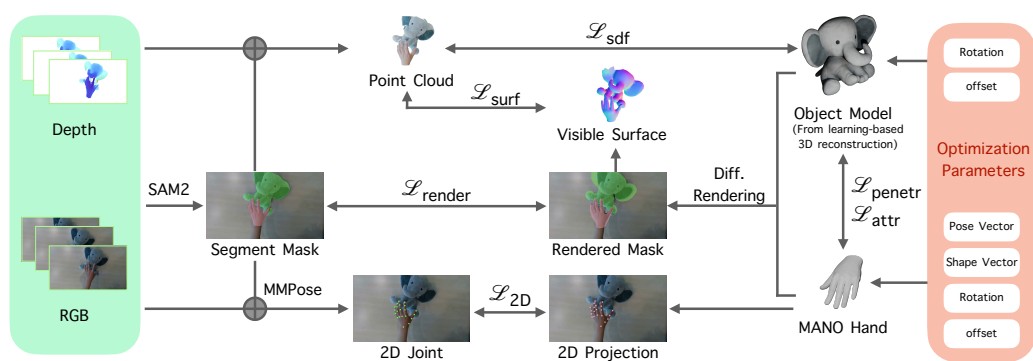

Figure 1: **Pipeline of TF-HOT**. Given an RGB-D video as input, we first use SAM2 and MMPose for mask prediction and 2D joint localization. We then jointly optimize the MANO hand model parameters (Romero et al., 2022) and object poses by minimizing a multi-term loss function, incorporating 2D priors from foundation perception models, 3D point cloud observation, and regularization to encourage physically-plausible and stable results. Our pipeline enables effective hand-object pose estimation for in-the-wild scenarios.

straints (Xie et al., 2024), transformer-based architectures (Pavlakos et al., 2024), and signed distance fields (Qi et al., 2024).

**Imitation Learning for Dexterous Manipulation**   Imitation learning (IL) is a promising paradigm for learning robot manipulation policies from expert demonstrations. The simplest form, Behavior Cloning (BC), learns policies to mimic expert actions from offline expert demonstrations and has been shown effective on a wide range of problems (Bain & Sammut, 1995; Pomerleau, 1988; Florence et al., 2022). However, BC struggles with out-of-distribution samples (Ross et al., 2011). Another approach to imitation is Inverse Reinforcement Learning (IRL) (Abbeel & Ng, 2004; Haldar et al., 2023), which focuses on learning to estimate expert reward function through online interactions, but it suffers from low sample efficiency (Kostrikov et al., 2018; Shen et al., 2022). Incorporating external data into Reinforcement Learning (RL) algorithms provides another approach to solving dexterous manipulation tasks, such as augmenting the replay buffers of RL (Vecerik et al., 2017; Radosavovic et al., 2021; Qin et al., 2022), or further incorporating additional behavior cloning loss terms (Rajeswaran et al., 2017). However, such methods typically rely on state-action demonstrations, which may not always be feasible when learning directly from human demonstrations (e.g., videos). Another method of incorporating expert demonstrations into RL is using state-only demonstrations to shape the reward (Xu et al., 2023; Christen et al., 2022; Wan et al., 2023). In our work, we utilize pose-only demonstrations from in-the-wild RGBD videos to optimize policies for dexterous manipulation tasks.

## 3    METHOD

Given an RGB-D video as input, the objective is to estimate the hand and object poses in each frame. As depicted in Fig. 1, we introduce a Training-Free 3D Hand and Object pose joint optimization pipeline (TF-HOT).

We use the MANO (Romero et al., 2022) model to represent the 3D hand shape. The MANO model provides two core functions, $\mathbf{J}$ and $\mathbf{M}$, which, given the input parameters of pose $\theta$, shape $\beta$, rotation $r$, and translation $t$, yield the hand mesh $\mathcal{M}$ and 3D hand joints $j^{\text{3d}}$. For brevity, we define the set of hand-related parameters as $\gamma = \{\theta, \beta, r, t\}$.

For object pose estimation, we assume that the object model $\mathcal{M}^{\text{obj}}$ is readily available. In practice, these models can be obtained through learning-based 3D reconstruction methods from single-view or multi-view images (Wei et al., 2024; Liu et al., 2024a;b; Xu et al., 2024; Hong et al., 2023). During the optimization process, the object pose $P$ for each frame is parameterized as a quaternion and a translation vector.

The overall goal is to jointly optimize the hand and object pose $\{\gamma, P\}$ in a per-frame manner by minimizing the following loss function:

$$
\begin{aligned}
\mathcal{L}_{\text{total}}(\gamma, P) = {} & \lambda_{\text{2d}}\mathcal{L}_{\text{2d}}(\gamma) + \lambda_{\text{render}}\mathcal{L}_{\text{render}}(\gamma, P) \\
& + \lambda_{\text{surf}}\mathcal{L}_{\text{surf}}(\gamma, P) + \lambda_{\text{sdf}}\mathcal{L}_{\text{sdf}}(P) + \lambda_{\text{penetr}}\mathcal{L}_{\text{penetr}}(\gamma, P) \\
& + \lambda_{\text{attr}}\mathcal{L}_{\text{attr}}(\gamma, P) + \lambda_{\text{reg}}\mathcal{L}_{\text{reg}}(\gamma, P)
\end{aligned}
\tag{1}
$$

Here, the $\lambda$ values serve as weighting coefficients for each loss term, and the loss terms are explained in the subsequent sections.

These loss components can be categorized into three groups: 1) 2D constraints applied in the image space; 2) 3D information that enhances pose accuracy; and 3) regularization terms that promote optimization stability and ensure physically plausible results.

## 3.1 Constraints from 2D Priors

**2D Joint Projection Loss $\mathcal{L}_{\text{2d}}$** We penalize the 2D projection error by measuring the Euclidean distance between projected 3D hand joints and reference 2D joint locations. This penalty is formulated as follows:

$$
\mathcal{L}_{\text{2d}}(\gamma) = \tilde{w}\|\Pi\mathbf{J}(\gamma) - \tilde{j}^{\text{2d}}\|^2
\tag{2}
$$

where $\mathbf{J}(\gamma)$ represents the MANO hand 3D joints, and $\Pi$ is the projection operator, $\tilde{j}^{2d}$ is the 2D joint locations predicted from the RGB images. The term $\tilde{w}$ corresponds to the 2D joint localization confidence. which adaptively modulates the loss weights to reflect the reliability of the joint predictions.

**Rendering Loss $\mathcal{L}_{\text{render}}$** To provide denser supervision, we employ a pixel-wise mask loss. Given the inherent hand-object interaction in our task, we jointly render both the hand and the object to account for occlusions:

$$
M^{\text{hand}}, M^{\text{obj}} = \pi[\mathbf{M}(\gamma), P_t\mathcal{M}^{\text{obj}}],
\tag{3}
$$

where $\pi$ is a differentiable mask renderer (Laine et al., 2020), $\mathbf{M}(\gamma)$ is the MANO hand model, $P_t\mathcal{M}^{\text{obj}}$ is the transformed object model using the object pose $P_t$.

We minimize the pixel-wise difference between the rendered masks and reference masks $\tilde{M}_t^{\text{hand}}, \tilde{M}_t^{\text{obj}}$:

$$
\mathcal{L}_{\text{render}} = w_1\|M^{\text{hand}} - \tilde{M}^{\text{hand}}\|^2 + w_2\|M^{\text{obj}} - \tilde{M}^{\text{obj}}\|^2,
\tag{4}
$$

where $w_1$ and $w_2$ are the respective weights for the hand and object.

In our implementation, we employ an off-the-shelf segmentation tracking network, SAM2 (Ravi et al., 2024), to obtain the reference hand and object masks. The hand masks are converted into bounding boxes, which are used as prompts for MMPose (Contributors, 2020) to predict 2d hand joints $\tilde{j}^{\text{2d}}$ along with their localization confidence $\tilde{w}$.

## 3.2 Leveraging 3D Information

In addition to the 2D constraints, we incorporate 3D information from the depth images to mitigate overfitting to the input view.

**Surface Loss $\mathcal{L}_{\text{surf}}$** The most widely used surface loss aligns meshes to point clouds by minimizing distance between point cloud and mesh surfaces such as in (Kwon et al., 2021). However, as shown in Fig. 2 (b), in single-view setting, we can only capture a partial point cloud, leading to ambiguity in determining which parts of the 3D model should align with the point cloud.

To address this problem, we introduce a **visible-aware surface loss**, which restricts alignment to the visible portion of the mesh surface. Since the visible faces are already computed during the rendering process, incorporating this loss does not introduce additional computational overhead.

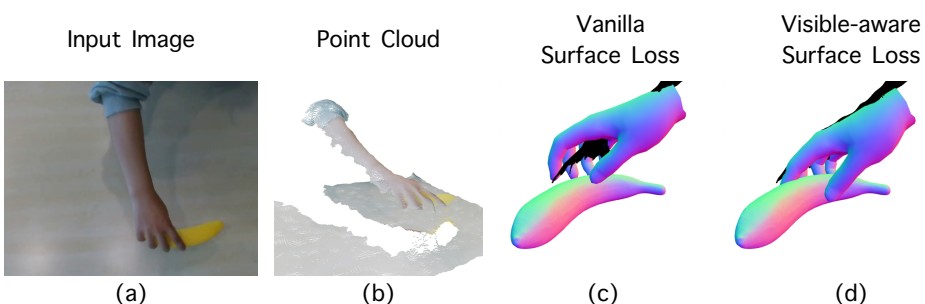

|            |            | Vanilla      | Visible-aware |
| Input Image | Point Cloud | Surface Loss | Surface Loss  |
| (a)        | (b)        | (c)          | (d)           |

Figure 2: **Illustration of the visible-aware surface loss**. (a) Input image; (b) Partial point cloud back-projected from the depth image (c) Vanilla 3D surface loss. The point cloud incorrectly fits the invisible surfaces of the hand model; (d) Visible-aware surface loss. The point cloud correctly aligns with the visible parts of the hand model. In (c) and (d), we only show the hand point cloud.

Formally, denoting the visible surface as $\mathcal{S}$, the visible-aware surface loss is defined as the combination of point-to-face and face-to-point distances:

$$f(\mathcal{P}, \mathcal{S}) = (w_3 \sum_{\triangle_i \in \mathcal{S}} \min_{p_j \in \mathcal{P}} \|p_j - \triangle_i\|^2 + w_4 \frac{|\mathcal{S}|}{|\mathcal{P}|} \sum_{p_i \in \mathcal{P}} \min_{\triangle_j \in \mathcal{S}} \|p_i - \triangle_j\|^2), \tag{5}$$

where $p_i$ is the $i$-th point in the point cloud $\mathcal{P}$, $\triangle_j$ is the $j$-th triangle of the visible surface, $\|p - \triangle\|^2$ computes point-to-triangle distance, $|\mathcal{S}|$ is the number of triangles in the visible surface, $|\mathcal{P}|$ is the number points in the point cloud, and $w_3$ and $w_4$ are the weights of point-to-face distance and face-to-point distance.

In our implementation, the hand and object masks are used to extract the respective point clouds, and the surface losses are computed for the hand and the object separately.

**SDF Loss $\mathcal{L}_{\mathbf{surf}}$**    The visible surface in the visible-aware surface loss is derived from the estimated object pose estimation rather than the ground truth. Consequently, if the object pose initialization is poor, the surface loss might attempt to align the incorrect parts of the surface to the point cloud, as shown in Fig. 3.

To address this problem, we follow (Chen et al., 2023) to use a Signed Distance Function (SDF) loss to minimize the distance between the point cloud and the surface defined by the zero-level set of the SDF field:

$$\mathcal{L}_{\text{sdf}}(P) = \sum_{v \in \mathcal{P}} \|\phi(P^{-1}v)\|^2, \tag{6}$$

where $\phi(x)$ is the trilinear interpolated SDF value at location $x$ from the object's SDF volume, $P^{-1}v$ represents the transformation of point $v$ back to the object's canonical space.

### 3.3 REGULARIZATION AND INITIALIZATION

In addition to the aforementioned 2D and 3D objective terms, We introduce several regularization terms to enhance the stability of the optimization process and ensure physically plausible results.

**Penetration Loss $\mathcal{L}_{\mathbf{penetr}}$**    A key physical constraint is to prevent hand-object intersection, as in (Hasson et al., 2019; Chen et al., 2023). This is enforced by penalizing the vertices of the hand that penetrated the object:

$$\mathcal{L}_{\text{penetr}}(\gamma, P) = \sum_{v \in \mathbf{M}(\gamma)} (-\mathbb{1}_{\phi(P^{-1}v)<0} \quad \phi(P^{-1}v)). \tag{7}$$

Instead of applying the maximum penalty as done in (Chen et al., 2023), we sum the penetration penalties across all hand vertices to impose a stricter non-penetration constraint.

**Attraction Loss $\mathcal{L}_{\mathbf{attr}}$**    Another physical constraint is the attraction loss, as proposed in (Hasson et al., 2019), which encourages contact between the fingertips and the object. This is achieved by penalizing the minimum SDF values of the five fingertips that are outside the object:

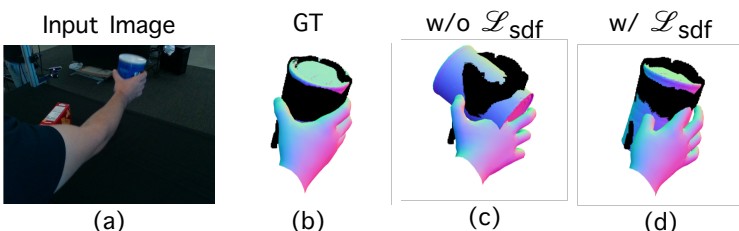

Figure 3: **Illustration of the SDF loss**. (a) Input image; (b) Ground-truth hand and object poses; (c) Result without the SDF loss. When the visible surface is inaccurate due to poor initialization, the visible-aware surface loss will guide the optimization in a wrong direction.; (d) Result with the SDF loss. It helps to converge to a better result with inaccurate initialization. Black points are object point cloud.

$$\mathcal{L}_{\text{attr}}(\gamma, P) = \sum_{i=i}^{n=5} \min_{v \in \mathbf{M}(\gamma)_C} (\mathbb{1}_{\phi(P^{-1}v)>0} \quad \phi(P^{-1}v)), \tag{8}$$

where $\mathbf{M}(\gamma)_{C_i}$ is predefined contact region of the i-th finger following (Hasson et al., 2019). We follow the same strategy as (Chen et al., 2023) to determine when the attraction loss should be applied: when the maximum penetration (i.e. the maximum of the negative SDF) exceeds a threshold, the hand is considered to be in contact with the object, and the attraction loss is activated to pull distant fingers closer.

**Regularization Loss $\mathcal{L}_{\text{reg}}$**  We penalize the difference of $j_t^{3d}, t_t$ across frames to stabilize results:

$$\mathcal{L}_{\text{reg}} = w_5 \max(0, \|j_t^{3d} - j_{t-1}^{3d}\|^2 - \epsilon_1) + w_6 \max(0, \|T_t - T_{t-1}\|^2 - \epsilon_2) \tag{9}$$

where $j_t^{3d}$ represents the 3D hand joints at frame $t$, $T_t$ is the translation component of object pose $P$ at frame $t$, $w_5$ and $w_6$ weights hand and object regularization, and $\epsilon_1$ and $\epsilon_2$ are predefined thresholds.

**Initialization**  Proper initialization is critical in tackling this high-dimensional optimization problem. In our implementation, we use different strategies for initializing the first frame and subsequent frames within a video.

For the first frame, we initialize the object pose using an off-the-shelf object pose estimation network (Wen et al., 2024). For the hand pose, we uniformly sample $N$ global hand rotations from the $\mathcal{SO}(3)$ manifold and randomly sample $N$ pose and shape parameters. The hand's global translations are initialized by aligning the center of the hand model $M_{\text{hand}}$ to the center of the hand point cloud. We then optimize $\gamma_0^i$ by minimizing $\mathcal{L}_{\text{total}}(\gamma_0^i, P_0)$ while keeping $P_0$ fixed, and select the optimal hand parameters $\gamma_0^j$ corresponding to the lowest 2D joint error: $j = \arg\min_i \mathcal{L}_{2d}(\gamma_0^i)$.

For subsequent frames, the optimized hand and object poses from the previous frame $(\gamma_{t-1}, P_{t-1})$ are used as the initialization for frame $t$.

### 3.4 Application: Pose Trajectory Following (PTF)

In this section, we demonstrate the application of our extracted hand and object poses in robotic dexterous hand manipulation tasks. We introduce an imitation learning method named Pose Trajectory Following (PTF) (Fig. 4), which leverages a single pose-only demonstration to optimize policies for dexterous hand manipulation tasks.

Given a trajectory of object and hand poses from TF-HOT, we apply inverse kinematics and the retargeting algorithm in (Qin et al., 2023) to set the initial robot joints angles such that the robot hand's initial pose and finger positions match the first frame of demonstration. To perform imitation learning using the pose-only demonstration, we design a specific trajectory-following reward (Tao et al., 2023) for dexterous hand manipulation (see supplementary materials for more details) that measures

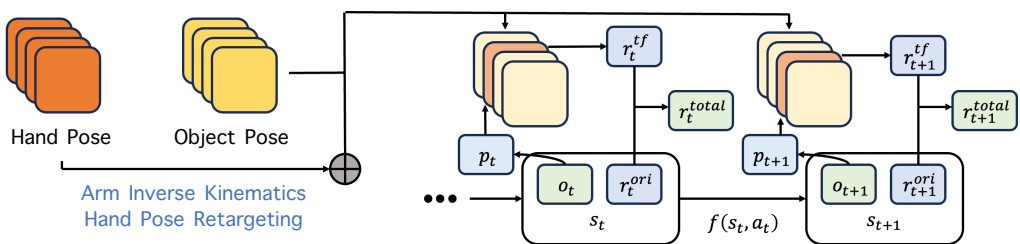

Figure 4: **Pipeline of PTF method.** The blue boxes contain the information used to calculate the final reward, including the poses ($p$), the original reward ($r^{ori}$), and the trajectory-following reward ($r^{tf}$). The green boxes contain the information provided to the agent for RL training.

the progress of the current robot hand's state along the target pose trajectory. We subsequently use PPO (Schulman et al., 2017) to optimize our policy to maximize the the total of trajectory-following return and origin return.

## 4 EXPERIMENTS

### 4.1 DATASETS AND METRICS

We conduct hand-object pose estimation experiments on two datasets: the public dataset DexYCB(Chao et al., 2021), a benchmark for hand-object interaction, and an in-the-wild dataset that we collected by ourselves using a RealSense D435 RGB-D camera.

DexYCB is a widely used real-world RGB-D dataset designed for hand-object interaction tasks, particularly object pick-up. It presents significant challenges due to its inclusion of fast hand motions and a variety of objects. We evaluate our method across four object categories—box, bottle, can, and bowl—using a total of 384 videos.

In our in-the-wild dataset, we captured 14 videos comprising 1,918 frames and featuring six distinct objects. Our method is evaluated across all available videos and frames.

For the DexYCB dataset, we use the following metrics for evaluation: 1) Hand pose evaluation: We report the mean per joint position error (MPJPE) and the pixel distance between projected 2D joints and their ground-truth locations (J2E). 2) Object pose evaluation: We measure the rotation error $r_{\text{err}}$ and the translation error $t_{\text{err}}$ for object pose estimation.

For the in-the-wild dataset, ground-truth labels are unavailable. Therefore, we evaluate hand and object pose estimation using the following proxy metrics: 1) Hand pose evaluation: We report the pixel distance between the projected 2D joints and the 2D joints predicted by MMPose, denoted as J2E*. 2) Object pose evaluation: We use the Intersection-over-Union (IoU) between the rendered object masks and the masks predicted by SAM2, denoted as $\text{IoU}_{\text{obj}}$. Additionally, we report the visible-aware 3D surface distance (defined in Eq. 5) normalized by the number of points in the point cloud, which we denote as $\text{SD}_{\text{obj}}$.

### 4.2 RESULTS AND ANALYSIS

We evaluate the performance of our method and compare it with two state-of-the-art approaches: HOTrack (Chen et al., 2023) and HOISDF (Qi et al., 2024). HOTrack utilizes an uncolored point cloud and 3D hand joints from the previous frame as input. It predicts the hand pose using a neural network, followed by a separate optimization module for both hand and object pose estimation. Since HOTrack requires the 3D hand joints and object pose from the first frame, we initialize the method with the first-frame results from our approach during the evaluation on our in-the-wild dataset. HOISDF only uses RGB images as input and employs a neural network to predict the per-point SDF value by aggregating image features. A subsequent module is used to estimate object poses guided by the SDF feature representations.

Table 1: **Quantitative results on hand and pose estimation**. We report MPJPE (cm), J2D (pixel), $t_{err}$ (cm), $r_{err}$ (°) for DexYCB and J2D* (pixel), IoU$_{obj}$, SD$_{obj}$ (mm) for in-the-wild data. * indicates that the results are compared with predictions from SAM2 and MMPose, while others are compared with ground-truth labels.

| | DexYCB | | | | In-the-wild | | |
|---|---|---|---|---|---|---|---|
| | MPJPE ↓ | J2D ↓ | $t_{err}$ ↓ | $r_{err}$ ↓ | J2D* ↓ | IoU$_{obj}$ * ↑ | SD$_{obj}$ ↓ |
| HOTrack (Chen et al., 2023) | 2.90 | 24.41 | 2.61 | **19.70** | 27.60 | 0.735 | 5.66 |
| HOISDF (Qi et al., 2024) | **1.01** | **5.02** | 2.92 | 40.10 | 25.33 | 0.258 | 16.0 |
| Ours | 2.56 | 11.14 | **2.39** | 30.12 | **6.68** | **0.786** | **5.59** |

Both of these two methods require training on large-scale annotated datasets, thus exhibiting worse generalization ability compared to our method.

**Evaluation on the DexYCB dataset**  We present evaluation results in Tab. 1. For hand pose estimation, our results outperform the in-the-wild tracking method HOTrack and are more physically plausible as illustrated in Fig. 5(a). While HOISDF achieves the best performance in terms of MPJPE and J2D, we argue that its superior results stem from dataset-specific priors embedded in its training process, which leads to overfitting on the DexYCB dataset and limits its ability to generalize. For object pose estimation, our method achieves the lowest translation error, which we attribute to the use of our visible-aware surface loss that enhances object localization accuracy.

**Evaluation on in-the-wild dataset**  When evaluated on the in-the-wild dataset, our method demonstrates superiority in both quantitative metrics (see Table 1) and qualitative outcomes (see Fig. 5(b)). HOTrack's reliance on point cloud input makes it sensitive to the quality of the point cloud, which, in this case, is derived from masks predicted by SAM2. The instability of the hand mask predictions negatively impacts HOTrack's performance. In contrast, HOISDF, which only utilizes RGB input and lacks direct access to 3D information, suffers from overfitting due to its reliance on learned dataset-specific priors. As a result, HOISDF struggles to generalize when confronted with data that includes unseen camera poses and objects not present in its training data.

### 4.3 ABLATION STUDY

We conducted ablation studies to analyze the impact of different loss components on the can category of the DexYCB dataset. Specifically, we compare the performance of our full pipeline with variations where each loss term is omitted. As shown in Tab. 2, the removal of any loss term leads to performance degradation, with the omission of the visible-aware 3D surface loss leading to particularly significant deviations.

We illustrate results in Fig. 6. The first row shows the input view and corresponding results, and the second row shows results from the back view perspective. Our full pipeline produces results that are both accurate and physically plausible, even in scenarios where the hand is significantly occluded. In contrast, when the visible-aware 3D surface loss is omitted, substantial misalignment occurs between the hand and the object. The removal of the penetration loss $\mathcal{L}_{penetr}$ results in hand-object penetration when viewed from the back, even though the input view appears correct. The absence of the attraction loss $\mathcal{L}_{attr}$ produces an unrealistic grasping posture, and without the regularization loss $\mathcal{L}_{reg}$, the hand becomes highly susceptible to noise in the depth data, leading to incorrect pose estimation results.

Table 2: **Ablation study on loss terms**. We report MPJPE (cm) evaluation results on the DexYCB can category.

| Ours | w/o visible | w/o $\mathcal{L}_{penetr}$ | w/o $\mathcal{L}_{attr}$ | w/o $\mathcal{L}_{reg}$ | w/o $\mathcal{L}_{sdf}$ |
|---|---|---|---|---|---|
| **2.86** | 4.45 | 3.71 | 3.73 | 3.16 | 3.60 |

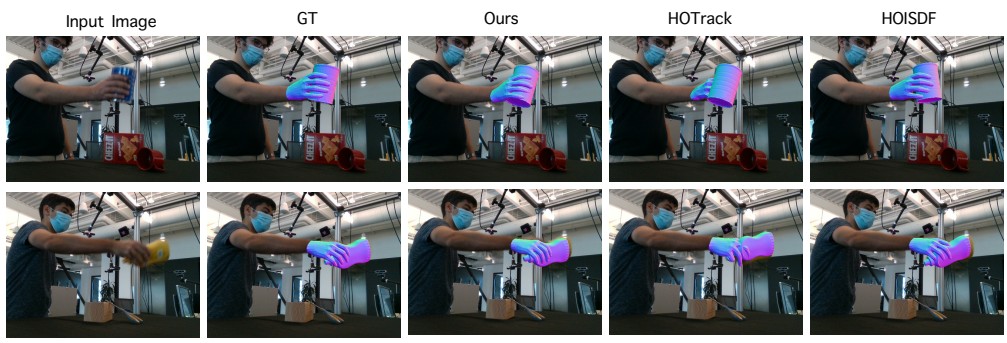

(a) DexYCB dataset.

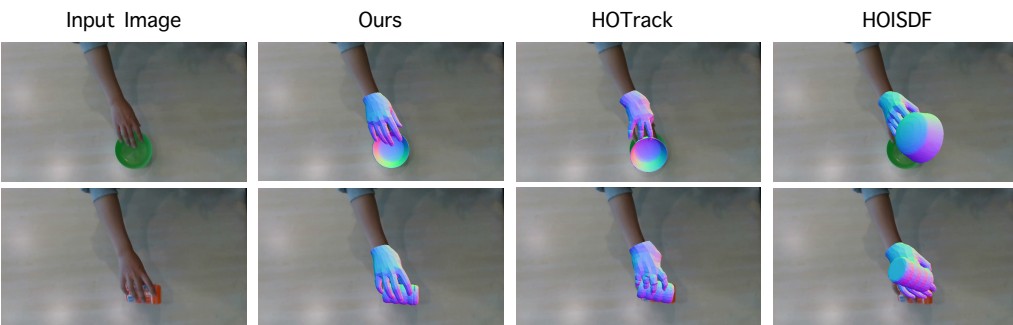

(b) In-the-wild dataset.

Figure 5: **Qualitative results on hand and object pose estimation**.

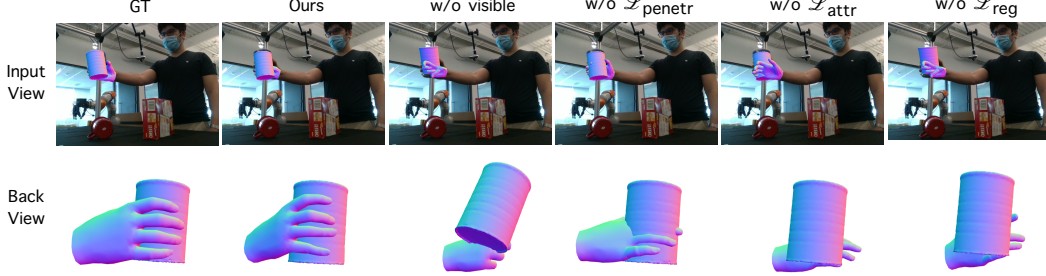

Figure 6: **Visualization of ablation study**. The first row shows hand and object estimation from the input view, the second row shows results from the back view.

### 4.4 APPLICATION

One of the roles of the demonstrations obtained from TF-HOT is to facilitate solving dexterous hand manipulation tasks using the PTF method. In this part, we conduct experiments to evaluate PTF using pose demonstrations obtained from TF-HOT and compare them against pure reinforcement learning methods and state-only imitation learning methods.

#### 4.4.1 EXPERIMENT SETUP

We focus on pickup tasks, a crucial component of dexterous hand manipulation. We conducted three tasks in the ManiSkill 3 (Tao et al., 2024). The robotic hand is the Inspire Hand, a 6-DoF robotic hand with five fingers. The initial states are shown in Fig. 7a. We add Gaussian noise to each joint of the robot and introduce positional perturbation to the object at the beginning of the task.

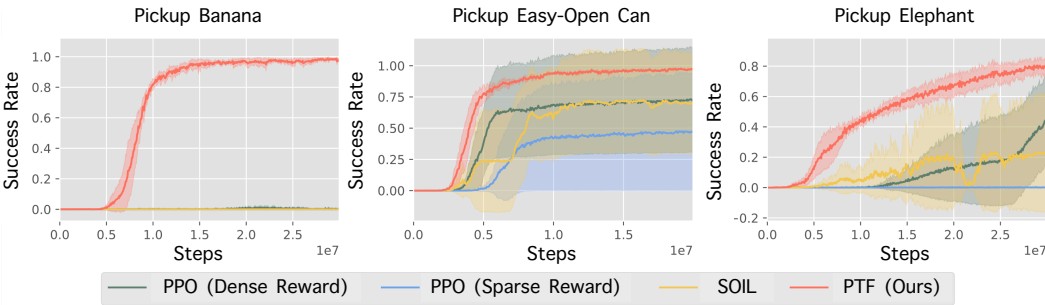

(a) Object models and initial states for the three tasks.  (b) Key moments during object grasping.

(c) Success rate curves over sample steps.

Figure 7: **Experiment results in Pickup tasks**. (a) shows the initial state of each task, (b) renders the results of the policy acting in the environment, and (c) plots the success rate curves over sample steps.

We compare PTF against two baseline methods: Proximal Policy Optimization (PPO) without the trajectory-following reward in PTF (but with the same robot initialization as PTF at the beginning of each episode), along with State-Only Imitation Learning (SOIL) (Radosavovic et al., 2021). We carefully design sparse and dense environment rewards for the baselines, with details in the supplementary material.

### 4.4.2 RESULTS

For each algorithm and task, we run four independent trials and report the average performance. The results are shown in Fig. 7c, where the solid line represents the average performance, and the shaded area indicates the variance across different random seeds.

As the results show, pure PPO with sparse rewards completely fails to pick up both the banana and the elephant. Although PPO with dense rewards perform better than those with sparse rewards, it also fails to successfully pick up the banana. For SOIL, the one-shot demonstration does not significantly improve its performance, as it achieves results similar to PPO. In contrast, our PTF method effectively utilizes the demonstrations, solving these tasks with a higher success rate and requiring fewer samples.

## 5 CONCLUSION

In this work, we proposed a Training-Free Hand and Object pose tracking framework (TF-HOT). Our method leverages differentiable rendering and rich priors from pre-trained 2D perception models for efficient optimization of human hand and object pose trajectories from input videos. We demonstrate that TF-HOT achieves superior performance over baseline methods on in-the-wild videos. Additionally, we showcase the application of our method in learning dexterous robotic tasks by introducing a Pose Trajectory Following (PTF) algorithm that trains a policy to follow the pose demonstrations extracted by TF-HOT from videos. Experiments demonstrate that our approach facilitates better and easier dexterous policy learning compared to reinforcement learning and imitation learning methods that do not utilize hand-object pose trajectory following.

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

## A   IMPLEMENTATION DETAILS

We use Adam as the optimizer with a learning rate of $lr = 0.01$. For the initial frame, we initialize $N = 20$ pars of $\{\theta_0^i, \beta_0^i, r_0^i, t_0^i\}$ hand parameters, optimize 500 iterations and choose the one with the best 2D Joint Project Loss. For the following frames, we optimize 250 iterations and set early stopping when the absolute difference of two sequential frames is less than 0.02 for DexYCB and 0.01 for in-the-wild data. We use (Wei et al., 2024; Liu et al., 2024b) to obtain object mesh from a single view image for in-the-wild data.

For DexYCB dataset, we use the same data processing process as HOTrack and use ground-truth segmentation masks as reference masks. For in-the-wild data, we use SAM(Kirillov et al., 2023) to manually segment the first frame hand and object mask. Additionally, we use per-frame object pose results from FoundationPose(Wen et al., 2024) as the initial pose estimation on in-the-wild data.

Hyper-parameters for hand-object pose estimation is as below:

|             | $\mathcal{L}_{2D}$ | $\mathcal{L}_{render}$ | $\mathcal{L}_{surf}$ | $\mathcal{L}_{sdf}$ | $\mathcal{L}_{penetr}$ | $\mathcal{L}_{attr}$ | $\mathcal{L}_{reg}$ |         |
|-------------|--------|---------|-------|------|---------|-------|-------|---------|
| DexYCB      | 10000  | 100     | 1     | 1    | 1       | 1     | 100   |         |
| in-the-wild | 300    | 100     | 0.1   | 0.1  | 0.1     | 1     | 100   |         |

|             | $w_1$ | $w_2$ | $w_3$ | $w_4$ | $w_5$ | $w_6$ | $\epsilon_1$ | $\epsilon_2$ |
|-------------|-------|-------|-------|-------|-------|-------|-----|------|
| DexYCB      | 1     | 1     | 1     | 1     | 1     | 1     | 0.02 | 0.03 |
| in-the-wild | 1     | 1     | 1     | 1     | 1     | 1     | 0.02 | 0.04 |

Table 3: **Hyperparameters for experiments of hand-object pose estimation**. Note that in the real implementation of $\mathcal{L}_{2D}$, projected 2D joints are normalized by image size.

## B   DETAIL OF POSE TRAJECTORY-FOLLOWING REWARD

In our method, two key trajectories need to be tracked: one is the pose of the hand's fingertips relative to the manipulated object, and the other is the absolute pose of the object in the world frame. The difference between two poses is measured by a function $d(p_1, p_2)$, where $p$ denotes the pose, consisting of both position and quaternion. We consider two poses to be matched when $d(p_1, p_2)$ is smaller than a constant $\epsilon$. At each step of the environment state, the trajectory-following reward is a metric that evaluates how much progress the state has made toward matching the given demonstration. Algorithms for computation of the trajectory-following reward during one episode rollout are described in Alg. 1. This reward function encourages the hand to achieve the correct posture relative to the object, while simultaneously guiding the object to the desired pose.

## C   APPLICATION EXPERIMENT DETAIL

### C.1   OBJECT RANDOMIZATION

To increase task difficulty, we introduce positional perturbations to the object in each task. Specifically, we apply a 5 cm perturbation to the banana in both directions in `Pickup Banana`, and a 1 cm perturbation in `Pickup Easy-Open Can` and `Pickup Elephant`.

### C.2   POSE DISTANCE FUNCTION

Although the pose distance function can be defined in various ways, we provide our specific definition here. We define the distance as

$$d(p_1, p_2) = \omega_1 ||\text{pos}_1 - \text{pos}_2||_2 + \omega_2 \cdot \texttt{diff\_rad}(\text{quat}_1, \text{quat}_2), \qquad (10)$$

where $\omega_1$ and $\omega_2$ are constants. `diff_rad` is a function that computes the angular difference between two quaternions.

---

**Algorithm 1** Pose Trajectory-Following Reward

---

**Input:** A sequence of poses $D$ with length $n$ as a demonstration ($D_i^{f_{1\sim5}}, D_i^o$ represent the pose of fingertips and object in the $i$-th frame, respectively).
**Output:** Trajectory-following reward $r^{tf}$ in the episode.
  # Reset the environment and set the prefix tracking index to 0.
  `env.reset()`
  $PT_{\text{obj}}, PT_{\text{hand}} \leftarrow 0$
  **repeat**
    $S \leftarrow$ Current environment poses.
    # $p_1 - p_2$ represents the pose of the first object relative to the second one.
    $T_{\text{hand}} \leftarrow$ The largest $i$ that $\sum_{j=1}^5 d(D_i^{f_j} - D_i^o, S^{f_j} - S^o) < \epsilon$
    $T_{\text{obj}} \leftarrow$ The largest $i$ that $d(D_i^o, S^o) < \epsilon$
    # Compute the trajectory-following reward, $\beta$ and $w$ are constants.
    $r_{\text{hand}}, r_{\text{obj}} \leftarrow 0$
    **if** $PT_{\text{hand}} < T_{\text{hand}}$ **then**
      $r_{\text{hand}} \leftarrow (1 + \beta \cdot T_{\text{hand}}) \cdot (1 - \tanh(w \cdot \sum_{j=1}^5 d(D_{T_{\text{hand}}}^{f_j} - D_{T_{\text{hand}}}^o, S^{f_j} - S^o)))$
    **end if**
    **if** $PT_{\text{obj}} = n$ or $T_{\text{obj}} = n$ **then**
      $r_{\text{obj}} \leftarrow 1 - \tanh(w \cdot d(D_n^o, S^o))$
    **else if** $PT_{\text{obj}} < T_{\text{obj}}$ **then**
      $r_{\text{obj}} \leftarrow (1 + \beta \cdot T_{\text{obj}}) \cdot (1 - \tanh(w \cdot d(D_{T_{\text{obj}}}^o, S^o)))$
    **end if**
    $r^{tf} \leftarrow r_{\text{hand}} + r_{\text{obj}}$
    # Update the prefix tracking index.
    $PT_{\text{obj}}, PT_{\text{hand}} \leftarrow \max(PT_{\text{obj}}, T_{\text{obj}}), \max(PT_{\text{hand}}, T_{\text{hand}})$
  **until** environment is terminal

---

## C.3   REWARD DESIGN

We design two types of environment rewards: sparse and dense. The sparse reward is a two-stage reward, where the agent receives 0.4 when the object is lifted and 1 when the object reaches the goal. The dense reward extends the sparse reward by adding terms for reaching. It can be written as:

$$r = \max \left\{ \text{Normalize} \left( w_1 r_{\text{hand reaching}} + w_2 r_{\text{obj reaching}} + w_3 \mathbb{I}_{\text{lifting}} \right), \mathbb{I}_{\text{success}} \right\}, \tag{11}$$

where $r_{\text{hand reaching}} = (1 - \tanh(c_1 \cdot d(\text{hand}, \text{obj})))$, $r_{\text{obj reaching}} = 1 - \tanh(c_2 \cdot d(\text{obj}, \text{goal}))$, and $w_1, w_2, w_3, c_1, c_2$ are constants. These terms are designed to encourage the hand to reach the object and lift it to the goal position.

## C.4   DEMONSTRATION COLLECTING

We recorded one video of a human hand picking up each object for each task. The video is then preprocessed by TFHO to extract the pose demonstrations for both the hand and the object. These demonstrations are used in the training of PTF and SOIL.

## C.5   HYPER-PARAMETERS

Here we provide the PTH method hyper-parameters for each task in Tab. 4.

# D   ADDITIONAL EXPERIMENTS OF THE MANIPULATION TASKS.

**Trajectory-Following Reward vs. Reaching Reward.** To visualize the impact of the trajectory-following reward on training, we selected the `Pickup Banana` and `Pickup Elephant` tasks and rendered one episode showing how policies trained with trajectory-following reward (PTF method) and reaching reward (PPO method with dense reward) behave in the environment. We captured the moments of object grasping, as shown in Fig. 7b.

| | Banana | Easy-Open Can | Elephant |
|---|---|---|---|
| num_envs | 512 | 512 | 512 |
| episode_length | 120 | 120 | 120 |
| sim_freq | 300 | 300 | 300 |
| origin_reward_scale | 5 | 5 | 5 |
| total_reward_scale | 0.01 | 0.01 | 0.01 |
| $\epsilon$ | 0.04 | 0.04 | 0.04 |
| $\beta$ | 0.1 | 0.1 | 0.1 |
| $w$ | 2 | 2 | 2 |

Table 4: **Training hyper-parameters for PTF method.**

We observed that policies trained with the reaching reward don't focus much on the hand's relative position to the object during grasping. For instance, in the banana pickup task, the hand curls its fingers before fully approaching the object, which hinders successful grasping. Similarly, in the elephant pickup task, the hand grasps the toy elephant by the ear, deviating from the intended goal of grabbing the body. In contrast, when training with the trajectory-following reward, the hand consistently moves into the desired pose, making it much more effective for picking up objects. This explains why the PTF method outperforms other algorithms in these tasks.

## E   LIMITATIONS

Despite the strengths of our method, it encounters limitations in scenarios where the hand is completely occluded or absent in the point cloud, as it lacks sufficient 3D priors to estimate accurate positions. However, these challenges could be mitigated by extending our framework to a multi-camera setup, which would improve accuracy through the aggregation of losses from multiple viewpoints. Additionally, our method has the potential to be utilized for automatic data annotation, offering a valuable tool for future research and applications.

