# OpenReview forum: "TF-HOT: Training-Free Hand-Object Pose Tracking and Optimization for Dexterous Manipulation"
_ICLR.cc/2025/Conference — Submitted to ICLR 2025_

### Official Review · Reviewer_H6mV · 2024-10-20

**Soundness:** 3
**Presentation:** 1
**Contribution:** 2
**Rating:** 3
**Confidence:** 4

**Summary:**

This paper presents TF-HOT, a novel, training-free approach to hand-object pose tracking that leverages differentiable rendering and 2D perception models (MMPose and SAM2) to estimate hand-object pose trajectories from videos. The method eliminates the need for large annotated datasets to reduce training costs. The authors also include an application study in imitation learning, where extracted pose trajectories are used to train dexterous manipulation policies.

**Strengths:**

1. Training-free method: One of the primary strengths is that the proposed method eliminates the need for large-scale annotated datasets. This makes it more adaptable to diverse environments and scalable for different applications, which is a valuable contribution given the challenges associated with collecting and annotating large datasets for hand-object interactions.

2. Generalization: Because the method is training-free, it is less prone to performance degradation when tested on OOD data, such as in-the-wild videos. This enhances the applicability of the method in real-world scenarios where training data might not fully cover the diversity of possible conditions.

3. Clear applicability: The authors showcase the utility of the proposed method by applying it in imitation learning tasks. The pose trajectory following task designed using the proposed method demonstrates its effectiveness in learning dexterous manipulation policies, adding real-world relevance to the contribution.

**Weaknesses:**

However, despite these strengths, the manuscript is not currently at the standard expected for ICLR. There are several areas where significant rework is necessary:

1. Related works are not thoroughly discussed: While the authors discuss some relevant works in the Hand and Object Pose Estimation section, they do not sufficiently connect this work to previous contributions in the literature. In particular, the discussion on the relationship between the proposed method and prior works is lacking. The paper would benefit from a dedicated subsection in the related work section that discusses differentiable priors. There is a lot of work in the area of HOPE and dexterous manipulation/grasping which use differentiable rendering or differentiable physics as a prior. For instance, [1] also addresses the HOPE problem using differentiable rendering, and the lack of a comparison weakens the context for understanding the novelty of this work.

2. Methodology is poorly presented: The method section introduces numerous loss terms (in sections 3.1, 3.2, 3.3) without offering a high-level overview of the proposed approach. Figure 1 is referenced, but without an accompanying explanation, this makes it challenging for the reader to understand the method. A clear breakdown of the pipeline before diving into the loss functions would greatly enhance the clarity of the presentation.

3. Unfair experimental design: The experiment design introduces concerns about fairness, particularly regarding the in-the-wild setting. The proposed method requires MMPose predictions to compute $L_{\text{2D}}$ during optimization, but the baseline methods do not require this information for inference. Therefore, comparing performance using J2E as a metric creates an information leak, as the proposed method has access to reference information that the baselines do not. The same critique applies to the other two metrics, IOU_obj and SD_obj, which are similarly tied to the optimization objective. These metrics are not valid for a fair comparison because the baselines are at a disadvantage by not having access to this reference information during inference.

4. Minor writing issues: There are some minor writing issues that detract from the paper's professionalism. For example, $L_{\text{surf}}$ should be $L_{\text{sdf}}$ in line 244. Additionally, $f(\cdot, \cdot)$ in eq. 5 and the first $P$ in eq. 6 are referred to without proper definition. J2E in line 358, line 363 is written as J2D in Table 1.

[1] HandyPriors: Physically Consistent Perception of Hand-Object Interactions with Differentiable Priors, Zhang et al., ICRA 2024

**Questions:**

1. How sensitive is the system to the quality of depth data? Given that depth information plays a critical role in the optimization process, would noisy or incomplete depth data significantly degrade performance?

2. The method includes a large number of hyperparameters (e.g., $\lambda$ values and $w$ values). How difficult is it to tune these parameters, and how were they selected for the experiments? Could the authors provide more insight into the selection process for these hyperparameters, as presented in Table 3?

3. Could the authors provide a more detailed timing analysis comparing the proposed method with learning-based methods during inference? What is the FPS performance of the proposed method, and how does it compare to the baseline methods? This would help in understanding the practical applicability of TF-HOT in real-time systems.

---

### Official Review · Reviewer_7QwF · 2024-11-03

**Soundness:** 2
**Presentation:** 2
**Contribution:** 2
**Rating:** 3
**Confidence:** 3

**Summary:**

The paper presents an optimization-based approach for reconstructing hand-object interactions from RGBD videos. The proposed technique, TF-HOT, requires no training stages, instead leveraging off-the-shelf segmentation and 2D keypoint prediction models to infer 2D and 3D cues from the input videos. With the object model assumed to be available, the method bypasses the 3D object reconstruction stage, allowing a direct focus on estimating object and hand poses. Evaluations in in-the-wild settings demonstrate the advantages of this training-free approach. Additionally, the paper showcases an application of the reconstructed hand-object trajectories by training a policy via imitation learning.

**Strengths:**

The training-free nature of the proposed technique significantly helps generalization. The paper presents an ablation study on the optimization terms. It demonstrates that they are complementary to each other, collectively improving performance.

**Weaknesses:**

The paper’s contribution and focus are unclear. First, it extends the optimization pipeline of prior work, reusing most of the existing energy terms, but which of these optimization terms are novel? Second, it’s unclear whether the proposed imitation learning approach is intended as a new contribution or simply as a demonstration of the applicability of the estimated hand-object interaction trajectories. If it’s the former, the reinforcement learning baselines are not competitive.

The paper also makes strong claims, particularly regarding the in-the-wild setting. However, it assumes the object model is available, which is an unrealistic requirement. This seems to be the main difference between the proposed TF-HOT and the two hand-object tracking baselines—HOTrack and HOISDF—which HOISDF first estimates an object SDF before tracking. This difference may lead to an unfair comparison. Additionally, both baselines report performance on the HO3D dataset, which is missing in this paper.

**Questions:**

1- Why is the initialization of the hand pose and shape random? Similar to how the object pose is initialized using an off-the-shelf object pose estimation model, both the hand and object could be initialized for the entire sequence. Furthermore, the proposed approach does not ensure an accurate hand shape, potentially causing artificially high errors due to hand shape mismatches.

2- The paper does not follow the metrics used in the HOTrack or HOISDF studies, which would facilitate a direct comparison.

3- In the ablations, performance declines significantly when the temporal smoothness term is removed (see the penetrations in the right-most figure in Fig. 6). It’s somewhat counterintuitive that removing temporal smoothness would lead to such artifacts.

Finally, I believe it should be “visibility-aware” rather than “visible-aware.”

---

### Official Review · Reviewer_gBv7 · 2024-11-04

**Soundness:** 2
**Presentation:** 3
**Contribution:** 1
**Rating:** 3
**Confidence:** 5

**Summary:**

The paper focuses on leveraging 2D video hand tracking model on in-the-wild videos. Based on a trained 2D perception model, the proposed method does inference-time optimization of hand pose parameters with using both 3D point cloud representation and the 2D prior from pre-trained models. With the estimated 2D hand skeleton and the point cloud observations, the method is applied to achieve the MANO parametric model parameters via optimization. Moreover, the authors provide demonstration to the potential of the proposed method transformed to imitation learning for hand trajectory following for robotic tasks.

**Strengths:**

The method proposed in this work is basically an optimization strategy to get hand shape parameters from a RGB-D videos. Off-the-shelf hand pose estimation and 3D point cloud estimation are required but no extra training is needed. This method is thus flexible to be integrated with different perception models to convert the hand keypoint skeleton to hand pose parameters under certain parametric model, such as MANO.

**Weaknesses:**

1. The proposed method requires a strong assumption that the object shape model is available. This is typically not available for in-the-wild videos. Even with the modern object shape reconstruction models, achieving an accurate and clean point cloud representation of 3D object shape is still challenging, especially when only monocular RGB observation is given.
2. Given that additional models are used for perception, it should be fair to also include video-based hand pose estimation model into the comparison, e.g, HaMeR[1]. Lacking this long line of related works provide insufficient information for the readers to estimate the significance of the proposed method.
3. The lack of novelty is a main concern of mine when evaluating this paper. Given that the perception results are provided by borrowing existing models, such as SAM, the main developed techniques in this work are only in the optimization stage. However, the related optimization terms, such as penetration term, 2d projection terms etc, have been studied in previous works. I can hardly recognize enough novelty and new contribution in this work.
4. For the self-collected in-the-wild video datasets, no enough information is provided to help readers understand it. It makes difficulty to estimate the experimental advance of the proposed method on the benchmark.
5. I highly recommend the authors to add more video-based (RGB or RGB-D) hand pose estimation methods into the benchmarking comparison. This is necessary to show the significance of the proposed method, which is important for the recognition from the research community.

Reference:

[1] “**Reconstructing Hands in 3D with Transformers**", CVPR 2024

**Questions:**

Please see my comments in the previous section.

---

> ### Comment · Reviewer_gBv7 · 2024-11-29
>
> Because no rebuttal is provided by the authors, my concerns remain not addressed. I remain my original rating for the paper.

---

### Official Review · Reviewer_aDEY · 2024-11-04

**Soundness:** 2
**Presentation:** 3
**Contribution:** 2
**Rating:** 5
**Confidence:** 4

**Summary:**

The authors propose a method for reconstructing hand-object interaction trajectories. The method does not depend on training data due to the use of per-sample optimization and off-the-shelf models. Numerous losses from prior literature and off-the-shelf methods are used to guide the optimization process. Ablations for the used losses are provided. A part of the DexYCB as well as a self-collected dataset are used for evaluation. An application using reconstructed hand trajectories in a reinforcement learning setup and showing superior performance over baselines is given.

**Strengths:**

The paper's writing is clear.

The method does not rely on human-annotated datasets for its losses.

The method is intuitive.

The method can process any given object category (with the performance presumably depending on the used off-the-shelf models and the heuristic values such as loss weights).

The proposed PTF algorithm looks interesting and useful.

**Weaknesses:**

The method depends on depth and ground-truth object meshes, which limits its applicability.

Only a single video is used per object category for the evaluation of PTF, which is hardly enough for training any method robust. Did I misunderstand something?

The method requires ground-truth/human-annotated object masks for the first step to initialize the mask tracking.

Limited evaluation on only four DexYCB objects: why were the rest of the object categories withheld? This makes comparison to other work difficult.

The ablation study of the method is performed on a single object category, which is susceptible to bias and unlikely to adequately capture its strengths and weaknesses.

Evaluation on a self-collected dataset: why not use another dataset with readily available hand and object pose data and performance reports of numerous other baselines, such as FPHA ("First-Person Hand Action Benchmark with RGB-D Videos and 3D Hand Pose Annotations", CVPR 2018), H2O ("H2O: Two Hands Manipulating Objects for First Person Interaction Recognition") or ARCTIC ("ARCTIC: A Dataset for Dexterous Bimanual Hand-Object Manipulation", Fan et al., 2017)?

I am willing to raise my score if thorough evaluations of the approach showing its superior performance over more state-of-the-art baselines are provided.

**Questions:**

What are your own contributions among the losses used? It appears that all proposed losses have been introduced in prior work.

---

### Meta-Review · Area_Chair_Jczg · 2024-12-19

**Metareview:**

The paper proposes an optimization-centered approach for the reconstruction of hand-object interactions from RGBD videos. The introduced technique, TF-HOT, obviates the need for training phases and instead utilizes pre-existing segmentation and 2D keypoint prediction models to extract 2D and 3D cues from the input videos.  Evaluations in real-world, unconstrained settings highlight the merits of this training-free methodology. Moreover, the paper exhibits an application of the reconstructed hand-object trajectories by training a policy through imitation learning.
Upon a thorough examination of the reviewers' remarks, the authors' rebuttals, and the ensuing discussions, it has been concluded that this paper ought to be declined. The reviewers brought forth several significant concerns. From a methodological perspective, the approach exhibits limitations such as its substantial reliance on depth and ground-truth data, evaluation restricted to single videos, the requirement for specific initializations, and a narrow evaluation range confined to only a few DexYCB objects. There are also potential biases in the ablation study. Doubts were also raised regarding the novelty of the method, as it amalgamates existing techniques without introducing substantial innovation in the core. Additionally, the experimental design was flawed, incorporating the use of inequitable comparison metrics and inconsistent metric application in comparison to related studies. The authors' rebuttals and subsequent discussions did not satisfactorily address these concerns.

**Additional Comments On Reviewer Discussion:**

no rebuttal is provided by the authors

---

### Decision · Program_Chairs · 2025-01-22

Reject